Different roles of urinary light chains and serum light chains as potential biomarkers for monitoring disease activity in systemic lupus erythematosus

Jiang Jun 1
Zhao Jin 2
Liu Dan 1
Zhang Man zhangman@bjsjth.cn 1 2 3
1 Clinical Laboratory Medicine, Peking University Ninth School of Clinical Medicine , Beijing , China
2 Clinical Laboratory Medicine, Beijing Shijitan Hospital, Capital Medical University , Beijing , China
3 Beijing Key Laboratory of Urinary Cellular Molecular Diagnostics , Beijing , China
Acharya Pragyan
Electronic publication date: 2022 May 17
Publication date: 2022
Volume: 10
Electronic Location ID: e13385
Received 2022 Jan 4; Accepted 2022 Apr 15
Copyright: ©2022 Jiang et al.
Copyright year: 2022
Copyright holder: Jiang et al.
License: This is an open access article distributed under the terms of the Creative Commons Attribution License, which permits unrestricted use, distribution, reproduction and adaptation in any medium and for any purpose provided that it is properly attributed. For attribution, the original author(s), title, publication source (PeerJ) and either DOI or URL of the article must be cited.
License URL: https://creativecommons.org/licenses/by/4.0/

Keywords: Disease activity, Light chains, Proteomics, Systemic lupus erythematosus, Urine

Funding: Beijing Key Clinical Specialty Program 2020ZDZK2 This work was supported by Beijing Key Clinical Specialty Program (grant number: 2020ZDZK2). The funders had no role in study design, data collection and analysis, decision to publish, or preparation of the manuscript.

==============================
Objective

The assessment system for monitoring systemic lupus erythematosus (SLE) disease activity is complex and lacks reliable laboratory indicators. It is necessary to find rapid and noninvasive biomarkers. The aim of this study was to screen and identify the differentially expressed proteins in urine samples between active SLE and stable SLE and to further explore the expression of light chains.

Methods

First, we used a label-free quantitative proteomics approach to establish the urine protein expression profile of SLE, and then screened differentially expressed proteins. Subsequently, the expression of overall light chains was examined by immunofixation electrophoresis and immunoturbidimetric methods, respectively.

Results

Mass spectrometry data analysis found a total of 51 light chain peptides in the urinary protein expression spectrum, of which 27 light chain peptides were differentially expressed between the two groups. The largest difference was IGLV5-45 located in the variable region of the immunoglobulin Lambda light chain. The levels of urinary light chains and serum light chains were both significantly elevated in active SLE, and the levels of urinary light chains increased with the severity of disease activity.

Conclusions

The measurement of light chains would help to monitor SLE disease activity. Serum light chains had better discriminatory capacity than urinary light chains, while urine light chains were closely related to the severity of disease activity and could be used for dynamically monitoring the progress of disease activity.

Introduction

Systemic lupus erythematosus (SLE) is a serious disease with a long course of illness that requires lifelong monitoring. Due to the repeatedly relapsing–remitting course, the current treatment of SLE is mainly based on long-term drugs and cannot be completely cured yet (Stojan & Petri, 2018). Although SLE patients have entered a stable phase after clinical treatment, they still need to monitor disease activity and relapse at any time, and the timeliness of such monitoring is very important and necessary in the disease management. At present, the activity monitoring of SLE mainly relies on clinical manifestations, and there are few objective monitoring indicators. The Systemic Lupus Erythematosus Disease Activity Index 2000 (SLEDAI-2K) has proven to be a useful measure of global disease activity in the clinic, especially in predicting mortality, which categorizing the disease activity into mild, moderate, severe and Lupus Low Disease Activity State (LLDAS) (Uribe et al., 2004; Franklyn et al., 2016). To protect important organs and reduce drug side effects, physicians would adjust the therapeutic regimen according to the different states of disease activity. Traditional immunological indicators such as serum complement and anti-double-stranded DNA antibodies cannot completely reflect disease activity, and their sensitivity and specificity are also heterogeneous due to different detection methods and thresholds. Urine has the advantages of noninvasiveness, convenient collection, resampling, and high patient compliance. In healthy individuals, 70% of the urinary proteins and peptides originate from the kidney and the urinary tract, whereas the remaining 30% represent proteins filtered by the glomerulus (Decramer et al., 2008). Urine not only directly reflects urinary system information, but also indirectly reflects plasma protein information, so it can be used as an ideal biomarker for urogenital diseases and nonurogenital diseases.

With the maturity and development of proteomics research, urine proteomics has become an important research direction for many diseases (Gonzalez-Buitrago, Ferreira & Lorenzo, 2007; Thongboonkerd, 2008). Kwon et al. (2021) found that urine alpha-1-acid glycoprotein (ORM1) could accurately predict early LN based on proteomic technology, while urine HBD had an excellent accuracy in distinguishing proliferative LN from non-proliferative LN. Some studies (Davies et al., 2021; Brunner et al., 2017) have verified a panel of urinary proteins, namely lipocalin-like prostaglandin D synthase (LPGDS), transferrin (TRF), ceruloplasmin, monocyte chemoattractant protein 1 (MCP-1) and soluble vascular cell adhesion molecule-1 (sVCAM-1), could predict active lupus nephritis. Although an increasing number of novel urinary biomarkers are being discovered, none of them was ideal predictor of lupus activity (Capecchi et al., 2020). Most of these urinary biomarkers are in the scientific research stage with lots of limitations, such as unstable urine samples, complex sample processing and detection methods, no standardized evaluation methods and high testing costs. Therefore, there are still many difficulties and challenges for urinary biomarkers to be widely applied in clinical laboratories. At present, most urine proteomics studies on SLE focus on the early diagnosis, activity monitoring and prognostic evaluation of lupus nephritis (Suzuki et al., 2009; Smith et al., 2017), and research on urinary biomarkers for the early reflection of overall disease activity in SLE is still lacking.

Therefore, in order to realize real-time monitoring of SLE disease activity and further explore the mechanism of disease activity, we hope to screen urine markers to find biomarkers that are fast, noninvasive, easily repeated, with greater sensitivity and specificity to assist clinicians in monitoring lupus disease activities. In this study, the aim of this study was to research the changes in urine kappa light chains (U-kappa) and urine lambda light chains (U-lambda) in the two groups of SLE patients, explore the new application of routine laboratory indicators, and further provide some objective basis for clinicians to monitor disease activity in time. To visually illustrate our research ideas, combined with the production and function of light chains, we drew the flow chart to describe the entire mechanism and process of this study (Fig. 1).

Figure 1 Experimental procedure description of the study.

(A) The production and function of immunoglobulin light chains. (B) The experiment process design of the study. IFE, Immunofixation electrophoresis.

Materials and Methods

Study subjects and samples

First, the study was approved by the local ethics committee of Beijing Shijitan Hospital, Capital Medical University and was in accordance with the principles of the Declaration of Helsinki (Approval number: sjtkyll-lx-2021(55)). All participants received an explanation of their participation rights and signed a written informed consent form. Then, SLE patients were diagnosed according to the revised American College of Rheumatology (ACR) criteria of 1997, excluding acute or chronic infections, tumors, other autoimmune diseases and severe kidney dysfunction caused by other diseases. The disease grouping was based on the SLEDAI-2000, which classifies SLE patients into a stable group (SLEDAI-2K<5) and an active group (SLEDAI-2K ≥ 5). According to the score, SLE patients with different degrees of disease activity are further divided into mild group (5 ≤SLEDAI-2K<10), moderate group (10 ≤SLEDAI-2K<15) and severe group (SLEDAI-2K ≥15) (Dörner & Furie, 2019).

Urine samples were collected in the morning that stayed 6–8 h in the bladder, obtained 10 ml from the midstream clean urine (avoiding the menstrual period and contamination by feces, secretions, etc.), and were transferred to the laboratory within 2 h. Then, urine samples were centrifuged at 1,500 rpm for 5 min to remove cells and cell debris, and the supernatants were divided into aliquots and frozen at −80 °C until use. The subjects fasted for at least 8 h, and the blood samples were collected into vacutainers without anticoagulant. Serum samples were separated from blood by centrifugation at 3,500 rpm for 10 min and the supernatants were stored in aliquots frozen at −80 °C until use. Then all enrolled subjects and laboratory indicators were extracted from their medical records.

Mass spectrometry

Based on label-free quantitative proteomics, urine samples from five active SLE patients, five stable SLE patients and five healthy controls, respectively, were processed by a Q-Exactive mass spectrometer (Thermo Fisher Scientific, Waltham, MA, USA) in our studies. Before proteomic analysis, the dissolution buffer (7M urea, 2M thiourea, and 0.1% CHAPS) was added to each urine sample, and the total proteins were grind and extracted by ultrasonic cell disruptor. Approximately 10 µg of each sample were loaded and electrophresed by SDS-PAGE. The gel was cut into pieces and subsequently destained, washed, and vacuum dried. Then, 200 µl dithiothreitol (DTT, 10 mM) was added to reduce the proteins at 37 °C for 1 h. To alkylate the proteins, 200 µl iodoacetamide (IAA) was applied in each sample in 55 mM. The reaction mixture was incubated for 0.5 h in dark place. After that, they were rehydrated in 100 µl of working solution containing 0.01 µg/µl trypsin, and digested overnight at 37 °C by adding 100 µl of 25 mM NH4HCO3. After enzymatic hydrolysis, the supernatants were extracted with 2.5% trifluoroacetic acid (TFA) and 50% acetonitrile (ACN). Then, the supernatant was loaded to the C18 desalting column. Desalting was conducted by washing three times with 0.1% formic acid (FA). After that, peptides were eluted with 70% ACN and freeze-dried in vacuum.

The components obtained by high pH reverse phase separation were re-dissolved in 20 µl of 2% methanol, 0.1% formic acid solution. The sandwich method was adopted, and 10 µl of peptide samples were separated using a C18 column with the flow rate of 350 nl/min. The isolated peptide was detected by mass spectrometer. Mass spectral data were analyzed and screened by MaxQuant software and the resulting differentially expressed proteins were matched by searching the UniProt Human database. Proteome default parameters of Maxquant software were as follows: Homo sapiens as searching Species; Trypsin as digestive enzyme; Carboxyamidomethylation (C) as fixed modification; Oxidation (M) and Acetyl (N-terminal) as variable modifications. Acquisition error of Q-Exactive mass spectrometer: 15 ppm for precursor ions; 20 mmu for fragment ions; 2 as the maximum missed cleavage site. The default cutoff value was used as a twofold change, and P-value < 0.05 was considered to indicate a significant difference.

Immunofixation electrophoresis analysis

An immunofixation electrophoresis analyzer (HELENA SPIFE 3000, Helena Laboratories, Beaumont, TX, USA) and supporting kits were used to detect three urine samples and three serum samples among the SLE stable groups, SLE active groups and healthy control groups, respectively. The urine samples of the stable group and the healthy control group were concentrated by 10K Amicon Ultra Centrifugal Filter Devices (Millipore Corp, Burlington, MA, USA), and the loading concentration and the volume were adjusted to 1 g/l and 20 µl, respectively. The serum samples were diluted 1:3 for serum protein lanes (SP) and diluted 1:5 for immunofix lanes (G, A, M, κ, λ), and the loading volume was 17 µl. After electrophoresis, the fixative, anti- γ antiserum, anti- α antiserum, anti- μantiserum, anti- κ antiserum and anti- λ antiserum were added to the lanes on the surface of gels for immunofixation. Then the gels were acid violet stained, decolorized by citric acid destain and dried. The immunofixation electrophoresis patterns were scanned by a scanner (Canon Corp, Tokyo, Japan) .

Determination of immunoglobulin light chains

The light chain levels were measured by the turbidimetric immunoassay method using an automatic special protein analyzer (SIMENS BN II, Simens Corp, Munich, Germany). All experimental operations strictly followed the department’s SOP documents and instructions. We defined urine kappa light chains (U-kappa) >7.16 mg/L as High U-kappa, and named urine lambda light chains (U-lambda) >3.94 mg/L as High U-lambda. The reference ranges of serum kappa light chains (S-kappa) and serum lambda light chains (S-lambda) were 1.7−3.7 g/L and 0.9−2.1 g/L, respectively. Similarly, we defined serum kappa light chains (S-kappa) >3.7 g/L as High S-kappa, and named serum lambda light chains (S-lambda) >2.1 g/L as High S-lambda.

Statistical analyses

Statistical assays were performed using SPSS version 21.0 and GraphPad Prism 9.0 software. Continuous variables were tested for normality using the Shapiro–Wilk test. The data conforming to a normal distribution are expressed as the mean ± standard deviation, while data with a non-normal distribution are expressed as the median (the interquartile range). Student’s t-test was used to analyze differences between the continuous variables, and the Mann–Whitney U test was employed for others. The qualitative data were described by frequency and percentage, and the comparison between two groups was analyzed by chi-square test and Fisher’s exact test. Receiver operating characteristic (ROC) curves were plotted to assess the sensitivity and specificity. Two-tailed P-values < 0.05 were considered statistically significant.

Results

Baseline characteristics of the study population

A total of 60 SLE patients were enrolled in our study, with a mean age of 41.72 ± 12.51 years and a mean SLEDAI-2000 score of 7.23 ± 6.86. Among them, ninety-five percent of the patients were females. There were 31 patients in the active group and 29 patients in the stable group. There was no significant difference in age(41.42 ± 12.12 vs 42.03 ± 13.13), sex (93.55% vs 96.55%) or disease duration (2.00 (0.42–7.00) vs 3.00 (0.92–5.50)) between the two groups. In active SLE, the disease activity was mild, moderate, and severe, accounting for 29.03%, 41.94%, and 29.03%, respectively.

Screening and identification of the expression of urine differential proteins

Differentially expressed proteins in urine of active SLE and stable SLE obtained by label-free technology. A total of 962 were differentially expressed between the two groups. The default cutoff value was used as twofold change, and a P value < 0.05 was considered to indicate a significant difference. There were 51 light chain peptides, of which 19 light chain peptides were upregulated and 8 light chain peptides were downregulated among these differentially expressed protein peptides. Six kappa light chain peptides were upregulated and six were downregulated. The expression of 13 lambda light chain peptides was upregulated, and two peptides were downregulated. Among them, the fold change of IGLV5-45 was the highest among these differentially expressed upregulated proteins (Table 1).

Table 1 The differential expression of light chains peptides in urine between active SLE patients (N = 5) and stable SLE patients (N = 5) (fold change >2 or <1/2, P-value < 0.05) by label-free quantification analysis.

Uinprot-ID	Protein name	Gene name	Ratio	P-value	Form of expression	
Kappa light chain peptides	
A0A075B6R9_HUMAN	Immunoglobulin kappa variable 2D-24 (non-functional) (fragment)	IGKV2D-24	17.31	0.0002	up	
A0A0G2JQJ0_HUMAN	Immunoglobulin kappa variable 1D-8 (fragment)	IGKV1D-8	13.23	0.0006	up	
KV621_HUMAN	Immunoglobulin kappa variable 6-21	IGKV6-21	6.39	0.0214	up	
KV105_HUMAN	Immunoglobulin kappa variable 1-5	IGKV1-5	5.74	0.0051	up	
KV117_HUMAN	Immunoglobulin kappa variable 1-17	IGKV1-17	4.31	0.0007	up	
KVD39_HUMAN	Immunoglobulin kappa variable 1D-39	IGKV1D-39	2.07	0.0356	up	
A0A075B6H7_HUMAN	Immunoglobulin kappa variable 3-7 (non-functional) (fragment)	IGKV3-7	0.03	0.0209	down	
KV127_HUMAN	Immunoglobulin kappa variable 1-27	IGKV1-27	0.04	0.0005	down	
KVD29_HUMAN	Immunoglobulin kappa variable 2D-29	IGKV2D-29	0.05	0.0162	down	
A0A075B6S9_HUMAN	Immunoglobulin kappa variable 1-37 (non-functional) (fragment)	IGKV1D-37	0.06	0.0002	down	
KVD16_HUMAN	Immunoglobulin kappa variable 1D-16 (fragment)	IGKV1D-16	0.08	0.00002	down	
KV113_HUMAN	Immunoglobulin kappa variable 1-13	IGKV1-13	0.09	0.0009	down	
Lambda light chain peptides	
A0A087WSX0_HUMAN	Immunoglobulin lambda variable 5-45 (fragment)	IGLV5-45	314.44	0.0007	up	
LV746_HUMAN	Immunoglobulin lambda variable 7-46	IGLV7-46	12.54	0.0447	up	
LV325_HUMAN	Immunoglobulin lambda variable 3-25	IGLV3-25	7.74	0.00002	up	
LV211_HUMAN	Immunoglobulin lambda variable 2-11	IGLV2-11	6.78	0.0001	up	
LV949_HUMAN	Immunoglobulin lambda variable 9-49	IGLV9-49	5.49	0.0039	up	
LV657_HUMAN	Immunoglobulin lambda variable 6-57	IGLV6-57	5.41	0.00002	up	
LV861_HUMAN	Immunoglobulin lambda variable 8-61	IGLV8-61	4.78	0.0005	up	
LV151_HUMAN	Immunoglobulin lambda variable 1-51	IGLV1-51	4.59	0.00006	up	
LV147_HUMAN	Immunoglobulin lambda variable 1-47	IGLV1-47	4.27	0.0007	up	
LV327_HUMAN	Immunoglobulin lambda variable 3-27	IGLV3-27	3.99	0.0212	up	
IGLL5_HUMAN	Immunoglobulin lambda-like polypeptide 5	IGLL5	2.91	0.0020	up	
LV140_HUMAN	Immunoglobulin lambda variable 1-40	IGLV1-40	2.70	0.0017	up	
IGLC3_HUMAN	Immunoglobulin lambda constant 3	IGLC3	2.34	0.0001	up	
LV136_HUMAN	Immunoglobulin lambda variable 1-36	IGLV1-36	0.29	0.0020	down	
LV214_HUMAN	Immunoglobulin lambda variable 2-14	IGLV2-14	0.38	0.0368	down	

Comparison of urine light chains between active SLE and stable SLE

The urine immunofixation electrophoresis patterns of active SLE, stable SLE and healthy controls are shown in Fig. 2A. There were no obvious monoclonal immunoglobulin bands in any lanes of the three urine samples. However, the patterns of active SLE demonstrated the presence of concentrated staining bands in IgG lane, kappa lane, and lambda lane, respectively, suggesting that the expression of IgG and light chain in the urine of active SLE were increased, while the stable SLE and the control group were lower.

Figure 2 Comparison of urinary light chains between stable SLE and active SLE.

(A) Immunofixation electrophoresis patterns of patient’s urine. The levels of U-kappa (B, D) and U-lambda (C, E) among different SLE disease activity groups. ROC curves analysis of U-kappa (F) and U-lambda (G) to distinguish active SLE from stable SLE. PATIENT1, active SLE; PATIENT2, stable SLE; PATIENT3, healthy controls.

The levels of U-kappa and U-lambda in the active group were significantly higher than those in the stable group (Figs. 2B and 2C). For the active groups, the positive rates of High U-kappa and High U-lambda were 70.97% and 54.84%, respectively. A total of 17 patients had High U-kappa and High U-lambda at the same time and the disease activity was mostly moderate and severe. For the stable groups, High U-kappa was identified for 37.93%, and High U-lambda was found for 24.14%, with 7 patients having High U-kappa and High U-lambda at the same time. The positive rates of High U-kappa and High U-lambda were both significantly higher than those of the stable groups (Table 2).

Table 2 Comparison of urinary light chains in SLE patients (N = 60).

	Stable SLE (n = 29)	Active SLE (n = 31)	Mild-SLE
(n = 9)	Moderate-SLE (n = 13)	Severe-SLE (n = 9)	P-value	
						Stable vs Active	Stable vs Mild	Stable vs Moderate	Stable vs Severe	
High U-kappa
n (%)	11 (37.93%)	22 (70.97%)	4 (44.44%)	10 (76.92%)	8 (88.89)	0.010	0.727*	0.019	0.019*	
U-kappa
(mg/L)	7.16
(7.16–12.95)	18.10
(7.16–84.90)	7.16
(7.16–17.55)	18.10
(7.53–37.90)	103
(18.00–187.00)	0.003	0.712	0.016	<0.001	
High U-lambda
n (%)	7 (24.14%)	17 (54.84%)	2 (22.22%)	7 (53.85%)	8 (88.89%)	0.015	0.906*	0.082*	0.001*	
U-lambda
(mg/L)	3.94
(3.94–4.11)	4.80
(3.94–48.50)	3.94
(3.94–4.17)	5.42
(3.94–12.80)	49.40
(7.85–91.45)	0.007	0.836	0.043	<0.001	
Notes.

Data are presented as median and inter-quartile range. P values were determined by the Mann–Whitney U test and chi-square test for quantitative and qualitative data, respectively.

* P were determined by Fisher’s Exact test for qualitative variables. Statistical significance was considered at P < 0.05.

Comparing the levels and positive rates of urine light chains in SLE patients with different disease activity levels, the levels and positive rates of U-kappa and U-lambda were highest in the severe group, followed by the moderate group, mild group, and stable group, decreasing in a stepwise manner from severe to no activity (Figs. 2D and 2E). There were no significant differences in urine light chain levels or positive rates between the mild SLE and stable groups (P > 0.05). Additionally, a statistically significant difference was observed between the severe SLE and stable groups, as well as between the severe SLE and mild SLE groups (Table 2).

ROC analysis of urine light chains

ROC curves were established based on the urine light chain levels of 31 active SLE patients and 29 stable SLE patients. The areas under the curves(AUCs) of U-kappa and U-lambda were 0.713 (95% CI [0.582–0.844]) and 0.679 (95% CI [0.543–0.814]), respectively. U-kappa had better diagnostic value for predicting the disease activity of SLE, with a cutoff value of 7.75 mg/l, and its sensitivity and specificity were 70.97% and 68.97%, respectively (Fig. 2F). However, the results revealed that U-lambda showed a lower AUC, with a lower sensitivity (48.39%) and the slightly higher specificity(86.21%), indicating that it has poor discriminatory capacity (Fig. 2G).

Comparison of serum light chains between active SLE and stable SLE

The serum immunofixation electrophoresis patterns of the active SLE, stable SLE and healthy controls are shown in Fig. 3A. There were no obvious monoclonal immunoglobulin bands in any lanes of the three serum samples. However, the patterns demonstrated the presence of concentrated staining bands in the IgG lane, IgA lane, kappa lane, and lambda lane, respectively. The bands of active SLE were the strongest staining, indicating that the expression of IgG, IgA and light chain in the serum of active SLE patients was significantly higher than that in stable SLE or healthy controls.

Figure 3 Comparison of serum light chains between stable SLE and active SLE.

(A) Immunofixation electrophoresis patterns of patient’s serum. The levels of S-kappa (B, D) and S-lambda (C, E) among different SLE disease activity groups. ROC curves analysis of S-kappa (F) and S-lambda (G) to distinguish active SLE from stable SLE. PATIENT1, active SLE; PATIENT2, stable SLE; PATIENT3, healthy controls.

Table 3 Comparison of serum light chains in SLE patients (N = 29).

	Stable SLE (n = 16)	Active SLE (n = 13)	Mild-SLE (n = 5)	Moderate-SLE (n = 4)	Severe-SLE (n = 4)	P-value	
						Stable vs Active	Stable vs Mild	Stable vs Moderate	Stable vs Severe	
High S-kappa, n(%)	4 (25%)	9 (69.23%)	4 (80%)	3 (75%)	2 (50%)	0.027	0.047	0.101	0.549	
S-kappa (g/L)	3.11 ± 1.37	4.73 ± 2.33	4.43 ± 1.65	5.17 ± 3.29	4.67 ± 2.61	0.028	0.089	0.302	0.324	
High S-lambda, n(%)	4 (25%)	7 (53.85%)	3 (60)	3 (75%)	1 (25%)	0.143	0.28	0.101	1	
S-lambda (g/L)	1.82 ± 0.61	2.57 ± 1.03	2.32 ± 0.41	2.77 ± 1.24	2.69 ± 1.51	0.014	0.101	0.035	0.331	
Notes.

Data are presented as the means ± standard deviation. P values were determined by the Student’s t-test for quantitative data and the Fisher’s Exact test for others.

* P < 0.05 was considered statistically significant.

Among 13 active SLE, 9 patients had High S-kappa and 7 patients had High S-lambda, and their disease activity was mostly mild and moderate. Among 16 stable SLE patients, only 4 patients had High S-kappa and High S-lambda at the same time (Table 3). For the active SLE patients, the mean concentrations of S-kappa and S-lambda were 4.73 ± 2.33 and 2.57 ± 1.03, respectively, which were both significantly higher than those of the stable SLE patients (Figs. 3B and 3C). Additionally, the positive rates of High S-kappa were significantly higher in active SLE than in stable SLE (69.23% vs 25%).

Comparing the levels and positive rates of serum light chains in SLE patients with different disease activity levels, the levels and positive rates of S-kappa and S-lambda were highest in moderate SLE. The levels of S-lambda were significantly higher in moderate SLE than in stable SLE (2.77 ± 1.24 vs 1.82 ± 0.61) (Figs. 3D and 3E). Similarly, the positive rates of S-kappa were significantly higher in mild SLE than in stable SLE (80% vs 25%) (Table 3).

ROC analysis of serum light chains

ROC curves were established based on the serum light chain levels of 13 active SLE patients and 16 stable SLE patients. Both S-kappa and S-lambda showed a high discriminatory capacity for disease activity, with AUCs of 0.721 (95% CI [0.534–0.908]) and 0.769 (95% CI [0.596–0.943]), respectively(Fig. 3F). The S-lambda cutoff point with the better diagnostic validity for disease activity of SLE was 1.95 g/L, with 76.92% sensitivity and 75.00% specificity (Fig. 3G).

Discussion

At present, the assessment of SLE disease activity is mainly based on clinical manifestations, which is complex and cumbersome. There are few indicators can be used to monitor the overall disease activity of SLE. To identify fast, convenient, and noninvasive markers for monitoring, we firstly used label-free quantification proteomic methodology to screen and identify potential urinary peptides from active SLE compared with stable SLE. The comparative data analysis revealed that many light chain peptides were differentially expressed and most of these peptides were in the variable domain of the immunoglobulins. During the adaptive immune response, the variable domain of immunoglobulins participates in antigen recognition, which was created by means of a complex series of V, D, and J gene rearrangement events and could then be subjected to somatic hypermutation after exposure to antigen to allow affinity maturation (Schroeder & Cavacini, 2010; Lefranc, 2014). These light chain peptides may have a potential role in predicting lupus activity. Here, we focused on the expression of the light chains in SLE patients.

The light chains are synthesized by B cells and have two isotypes, kappa and lambda, with monoclonal and polyclonal properties. Monoclonal light chains are produced by the malignant proliferation of a single B cell clone, and were initially used to diagnose and monitor plasma cell dysplasia, such as multiple myeloma, non-Hodgkin’s lymphoma and other blood system diseases (Rajkumar et al., 2014). Polyclonal light chains are usually produced by multiple B cell clones due to chronic immune stimulation. In recent years, several studies have demonstrated that the light chain might act as a biomarker for monitoring the activation of B cells in autoimmune diseases and chronic inflammation, and serum light chain concentrations are related to the activity or recurrence of some autoimmune diseases (Basile et al., 2017; Deng et al., 2015; Lanteri et al., 2014; Verstappen et al., 2018). SLE has a complex etiology, involving the interaction of multiple factors such as heredity, environment and hormones, and is characterized by the production of different autoantibodies and inflammatory cytokines, which ultimately lead to immune disorders and cause organ damage, while B cell activation plays an important role in the etiopathogenesis of SLE (Herrada et al., 2019; Moulton et al., 2017). During periods of SLE disease activity, polyclonal B cells are hyperactive and synthesize large amounts of immunoglobulins accompanied by excessive light chain synthesis. Approximately 60% of those excessive light chains are incorporated into the immunoglobulin complex and the remaining 40% circulate in the free form (Waldmann, Strober & Mogielnicki, 1972). Moreover, the lambda light chains are released as dimers, and the kappa light chaina are released as monomers or dimers (Bradwell, 2005; Boivin et al., 2004). In addition, some studies have found that free light chains interact with mast cells and neutrophils, with the potential to promote inflammation (Thio et al., 2008; Redegeld et al., 2002; Bramlage et al., 2016).Therefore, the concentrations of free light chains could reflect the state of inflammation.

In recent years, several studies have shown that the concentrations of serum free light chain were significantly higher for patients with SLE disease activity or relapse than for stable SLE patients, which may be useful biomarkers for monitoring lupus activity (Draborg et al., 2016; Mastroianni-Kirsztajn, Nishida & Pereira, 2008). However, few studies have shown that there is a high correlation between the levels of urine light chains and disease activity. Hopper et al. observed that the concentrations of urine light chains in 20 SLE patients were significantly increased during active phases and reached normal values only after remission (Hopper et al., 2000; Hopper et al., 1989). These results suggested that the levels of urine light chains might be markers of polyclonal B cell activity in SLE. Hanaoka et al. found that urinary levels of light chains were significantly higher for patients with proliferative lupus nephritis(ISN/RPS class III/IV) than for patients with non-proliferative lupus nephritis(ISN/RPS class I/II/V), which might be useful biomarkers in proliferative lupus nephritis (Hanaoka et al., 2013). However, these early studies mostly focused on patients with lupus nephritis, and the sample size was small with radioactive methods. Therefore, further studies are needed to explore the expression of urine light chains during SLE active phases and the relationship to different degrees of disease activity before the urine light chains can be used as ideal biomarkers for monitoring disease activity.

This study evaluated the levels of overall light chain in the urine and serum of SLE patients with qualitative and quantitative methods. Our results showed that the expression of light chains in urine and serum of active SLE were significantly increased, and their ratio (Kappa/Lambda) was normal, indicating polyclonal rather than monoclonal elevation. The patterns of immunofixation electrophoresis were consistent with the results of the immunoturbidimetric assay. The strong association of the levels of urine light chains with different disease severities or disease activities of SLE, and the expression of urine light chains increased with increased lupus activity. Among them, the levels of urine light chains were highest in severe SLE. Therefore, the levels of urine light chains reflected the B cell activity and the degree of inflammation in SLE patients to a certain extent, which might be used as indicators to monitor the aggravation of lupus activity or evaluate the clinical effect.

In addition, our study also found that the expression of light chains in serum was the highest in moderate SLE, but only the levels of S-lambda were statistically significant between moderate SLE and stable SLE. In contrast, only the positive rates of S-kappa were statistically significant between mild SLE and stable SLE. This difference may indicate that S-kappa could be used to monitor the early stage of SLE disease activity, but it is more meaningful to monitor the changes in S-lambda as the disease activity increases. Interestingly, the levels of S-kappa and S-lambda both decreased during the severe period of SLE disease activity. These results are similar to those reported by Cambron et al. (2020), who compared the levels of light chains between nonflare SLE patients and moderate/severe flare SLE patients, observing only a significant increase in the expression of S-lambda. Therefore, the levels of serum light chains might be used to identify or predict lupus activity in the early and moderate phases of SLE, but this assumption needs more evidence to support by expanding the sample size.

Another result of interest is the discriminatory capacity for disease activity of light chains, and our results indicated that U-kappa, S-kappa and S-lambda all have good discriminatory capacity. For S-kappa and S-lambda, the cutoff values were consistent with the reference range of in our laboratory. Similarly, our results showed that S-Lambda had a better discriminatory capacity, which was in agreement with those reported by Cambron et al.

On the basis of proteomics technology, we firstly screened urine differentially expressed proteins of active lupus in the identification phase. Subsequently, in order to explain the relationship between light chain peptides and lupus activity, we examined the expression of the overall light chains in urine and serum, respectively, further analyzing their relationship with different degrees of disease activity in the verification phase. These light chains were detected from the local to the whole, qualitative to quantitative. Finally our results indicated that light chains had discriminatory capacity for the presence of lupus activity and might be potential biomarkers for monitoring activity and severity.

In conclusion, the expression of light chains in urine and serum was increased in active SLE, which is closely related to different degrees of disease activity, indicating B cell hyperactivity and an inflammatory state in SLE patients. Combined detection of the levels of light chains in urine and serum would help to monitor lupus activity, and urine light chains could be used as potential markers of disease severity. Further longitudinal studies with large-scale samples are needed to evaluate the clinical value of the two isotypes of light chains in monitoring disease activity and analyze their changes in different activity degrees and damaged organs.

Supplemental Information

Supplemental Information 1 Results of mass spectrometry for the differentially expressed light chain peptides in SLE

‘ID’, ‘Accession’, ‘Gene Name’ and ‘Description’, all four are used to describe differentially expressed light chain peptides. ‘Sig(active_stable)’ represents the protein differential expression between the active SLE group and stable SLE group. ‘0’ means no difference, ‘1’ means up-regulated protein expression, and ‘−1’ means down-regulated protein expression ‘Ratio(active_stable)’ represents fold change of the protein differential expression between the active SLE group and stable SLE group. Columns E to P respectively represent the relative expression of the differentially expressed peptides detected in three replicates of mass spectrometry in SLE patients and control groups.

Click here for additional data file.

Supplemental Information 2 Results of urinary light chains in SLE patients by turbidimetric immunoassay

Click here for additional data file.

Supplemental Information 3 Results of serum light chains in SLE patients by turbidimetric immunoassay

Click here for additional data file.

Additional Information and Declarations

Competing Interests

Author Contributions

Human Ethics

Data Availability

The authors declare there are no competing interests.

Jun Jiang conceived and designed the experiments, performed the experiments, analyzed the data, prepared figures and/or tables, authored or reviewed drafts of the paper, and approved the final draft.

Jin Zhao performed the experiments, analyzed the data, authored or reviewed drafts of the paper, and approved the final draft.

Dan Liu analyzed the data, prepared figures and/or tables, and approved the final draft.

Man Zhang conceived and designed the experiments, prepared figures and/or tables, authored or reviewed drafts of the paper, and approved the final draft.

The following information was supplied relating to ethical approvals (i.e., approving body and any reference numbers):

This study was approved by the local ethics committee of Beijing Shijitan Hospital, Capital Medical University (Ethical Application Ref: sjtkyll-lx-2021(55)).

The following information was supplied regarding data availability:

The raw measurements are available in the Supplementary Files.

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
