# Peer review of "Different roles of urinary light chains and serum light chains as potential biomarkers for monitoring disease activity in systemic lupus erythematosus"

_PeerJ, doi:10.7717/peerj.13385_

## Round 0.1 · original submission · Major Revisions

The manuscript seems to have major problems with the definitions used which are not current and the data analysis. Major revision is recommended for this manuscript. ALL of the reviewers' concerns must be addressed before the manuscript is considered for further review.

Reviewer 1 ·

Basic reporting

Thank you for providing the Table 1 Label free quantification analysis. However, the data needs to be supported by providing the raw files generated from Q-Exactive mass spectrometer.

Experimental design

I appreciate the authors extensive effort to compile human SLE data set. The manuscript is well written with professional language. Necessary ethical approval statement is provided. The identifiable patient information has been removed.


In Line 97 section 1.2 Mass spectrometry, the authors need to describe in brief how they processed their urine samples for Label free quantitative proteomics. What was the enzyme used, was reduction/alkylation performed, were the samples desalted etc before processing them on Q-Exactive mass spectrometer?

While the authors have included healthy group as control for immunofixation electrophoresis analysis, however in Line 98-99, the authors have only used active and stable SLE patients for proteomics analysis without controls. The control samples should be included.

Validity of the findings

In the discussion section line 290 – 293, the authors mention using proteomics technology to examine the expression of the overall light chains in urine and serum. However, there is no mention of processing serum samples in the Mass Spectrometry section 1.2. The authors need to clarify if serum samples were processed for label free proteomics analysis.

Reviewer 2 ·

Basic reporting

1. The introduction has been written in a vague manner. Instead of elaborating on the disease activity parameters and symptom flare, the authors should describe the limitations of hitherto known urinary biomarkers. The part: 'To protect important organs and ........... accompanied by irreversible organ damage' is redundant and irrelevant. A concise note on the urinary biomarkers would help readers understand the current scenario about them in SLE.

Experimental design

The experimental design can not be commented upon unless the below-mentioned 3 parameters are addressed adequately.

1. Why have the authors chosen to use the 1997 Lupus classification criteria which have now been replaced by the 2012 SLICC and 2019 ACR/EULAR criteria?

2. Dividing patients' severity status just based on SLEDAI is incorrect. They also need to see that renal SLEDAI (rSLEDAI) which is a sum of all the 4 renal parameters of SLEDAI is also taken into account. For example, a patient with arthritis, skin rash, and raised anti-ds DNA/low complements would score >5 on SLEDAI but can not be called a severe disease whereas a patient with nephrotic range proteinuria with raised anti-ds DNA/low complements would also fall into the same group but would be classified as severe disease as he/she would need more intense immunosuppression as compared to the previous one. This is a major drawback of the study where patients with and without renal involvement have been grouped together on the basis of the same SLEDAI value which is likely to give incorrect results. For this kind of study, it is imperative that patients of the same kind i.e. renal active OR non-renal active be classified separately into mild, moderate, and high disease activity.

3. How have the authors defined active and stable disease? If they have used SLEDAI, what's the definition of active disease as per the SLEDAI value?

Validity of the findings

The validity of the findings would further depend upon the case definition which needs to be elaborated as per my comments in the above section.

Reviewer 3 ·

Basic reporting

In the article "Different roles of urinary light chains and serum light chains as potential biomarkers for monitoring disease activity in systemic lupus erythematosus", the authors investigated the changes of antibody light chain components in serum and urine in order to develop a more precise and diagnostic biomarker for systemic lupus erythematosus. Since SLE is known to be an auto-immune disease, the carried out study compared and contrast the antibody light chain components between active and stable SLE patients, and established a diagnosis guideline using these information at the end. The paper is clearly written and sufficiently referenced for readers to understand the urgency of the problem as well as the underlying mechanism and conclusion.

- Fig. 1A and Fig. 2A miss labels of corresponding category, mild, moderate, severe, or stable.
- Use “scatter plot with bar” graph in Prism to show the datapoint distribution for all bar graphs in the paper.
- Fig. 3 is better to be explained and presented at the beginning of the paper.

Experimental design

For the ten samples used for screening and identification of the urine proteins, the authors need to clarify whether/how N=5 was picked for each group and was representative for the group. Especially, which one category, mild, moderate or severe, was chosen for the active group?

Validity of the findings

The presented data from serum protein comparisons, although in some cases statistically significant, do not show distinguishable changes to be a reliable bio-marker among patients with different disease severity. Would “expanding the sample size” really make the changes more significant as the authors proposed? Or, it might just be a worse indicator compared to urine samples.

---

## Round 0.2 · accepted · Accept

The authors have addressed all the concerns of the reviewers and the manuscript is therefore ready for acceptance.

Reviewer 1 ·

Basic reporting

The authors have made the necessary changes as requested and have added more information as requested.

Experimental design

The authors have described the processing of Urine samples in more details compared to their previous draft of manuscript. The Mass Spectrometry data has been added as per request.

Validity of the findings

The urine and serum data has been validated well. The methodology as well as raw data support their findings.

Reviewer 3 ·

Basic reporting

no comment

Experimental design

no comment

Validity of the findings

no comment

Additional comments

The revision has addressed all my comments.